# Inhibition of c-Jun in AgRP neurons increases stress-induced anxiety and colitis susceptibility

Fuxin Jiao[1,2], Xiaoming Hu[1], Hanrui Yin[2], Feixiang Yuan[1], Ziheng Zhou[2], Wei Wu[3], Shanghai Chen[1], Zhanju Liu [3✉] & Feifan Guo [1✉]

Psychiatric disorders, such as anxiety, are associated with inflammatory bowel disease (IBD), however, the neural mechanisms regulating this comorbidity are unknown. Here, we show that hypothalamic agouti-related protein (AgRP) neuronal activity is suppressed under chronic restraint stress (CRS), a condition known to increase anxiety and colitis susceptibility. Consistently, chemogenic activation or inhibition of AgRP neurons reverses or mimics CRS-induced increase of anxiety-like behaviors and colitis susceptibility, respectively. Furthermore, CRS inhibits AgRP neuronal activity by suppressing the expression of c-Jun. Moreover, overexpression of *c-Jun* in these neurons protects against the CRS-induced effects, and knockdown of *c-Jun* in AgRP neurons (*c-Jun*$^{\Delta AgRP}$) promotes anxiety and colitis susceptibility. Finally, the levels of secreted protein thrombospondin 1 (THBS1) are negatively associated with increased anxiety and colitis, and supplementing recombinant THBS1 rescues colitis susceptibility in *c-Jun*$^{\Delta AgRP}$ mice. Taken together, these results reveal critical roles of hypothalamic AgRP neuron-derived c-Jun in orchestrating stress-induced anxiety and colitis susceptibility.

[1] Zhongshan Hospital, State Key Laboratory of Medical Neurobiology, Institute for Translational Brain Research, MOE Frontiers Center for Brain Science, Fudan University, Shanghai 200032, China. [2] CAS Key Laboratory of Nutrition, Metabolism and Food Safety, Innovation Center for Intervention of Chronic Disease and Promotion of Health, Shanghai Institute of Nutrition and Health, University of Chinese Academy of Sciences, Chinese Academy of Sciences, Shanghai 200031, China. [3] Department of Gastroenterology, The Shanghai Tenth People's Hospital, Tongji University, Shanghai 200072, China. ✉email: liuzhanju88@126.com; ffguo@fudan.edu.cn

Psychological disorders, such as anxiety and depression, severely influence human health[1]. What makes it even worse is that they are frequently associated with the occurrence of many other diseases, including inflammatory bowel disease (IBD)[2–4]. Stress is a common cause of psychiatric disorders[5] and intestinal inflammation[6,7]. Accumulating evidence have illustrated that brain-gut interactions play an important role in the outcome of psychological disorders and IBD under stress conditions[8–10]. Although gut signals and microbiota are proposed to play roles in the comorbidity of psychological disorders and intestinal inflammation[7], the neural mechanisms behind stress-induced anxiety and colitis susceptibility are unknown.

The hypothalamus is an important neural control center for regulating stress response[11], consisting of several important nuclei. The hypothalamic arcuate nucleus (ARC) is critical to the regulation of energy metabolism[12] and recently reported to be engaged in emotional regulation[3,13,14]. There are two specific populations of neurons in the ARC: neurons co-expressing the orexigenic neuropeptide agouti-related protein (AgRP) and neuropeptide Y, and neurons co-expressing the anorexigenic pro-opiomelanocortin (POMC) precursor and the cocaine- and amphetamine-related transcript. The AGRP/NPY neurons have an inhibitory effect on the POMC/CART neurons[15]. Additionally, AgRP expressing cells are limited to ARC[16] and its activity is influenced by stress[14]. AgRP neuronal activity controls diverse physiological processes, including feeding[17], pain sensation[18] and food-seeking behavior[17,19]. Studies shows that AgRP neurons are involved in the regulation of peripheral tissue homeostasis[20,21], suggesting that these neurons may play an important role in stress-induced intestinal disorders.

c-Jun is a component of the activator protein-1 transcription factor family, which forms a homodimer or heterodimer with other members of the family (c-Fos or ATF) to manipulate downstream target genes[22]. It is expressed in many tissues, including the brain[23]. c-Jun has been implicated in the regulation of brain-associated disease. Mice lacking c-Jun develop severe defects in axonal response after transection of the facial nerve[24]. Inhibition of c-Jun could attenuate axotomy-induced dopamine neurons death[25]. Current evidence has revealed that c-Jun is activated in the brain by chronic antidepressant treatment and controls the p11-dependent antidepressant response[26]. These findings indicate that c-Jun may be associated with psychiatric disorders. In addition, c-Jun is also involved in a variety of stress responses. It has been reported that c-Jun protects cells from stress-induced apoptosis[22] and is required for preventing stress-imposed maladaptive remodeling of heart[27]. Furthermore, c-Jun phosphorylation expression is controlled by chronic isolation stress in the prefrontal cortex[28]. Because c-Jun is an immediate-early gene that is dynamically regulated in response to neuronal activity[29], it is commonly used as a marker reflecting neuronal activity[30]. Considering c-Jun is induced under stress conditions[28] and highly expressed in the hypothalamus arcuate nucleus[31], where the AgRP neurons are enriched, suggesting that it may additionally be involved in the regulation of some important functions in the AgRP neurons.

Based on the above knowledge, we hypothesize that c-Jun in AgRP neurons plays an important role in the stress-induced comorbidity of anxiety and IBD. This study investigates such a possibility and explores the likely mechanisms.

## Results

**Chronic Restraint Stress (CRS) induces anxiety-like behaviors and increases susceptibility to colitis.** To induce anxiety and colitis, we employed a CRS mouse model (Supplementary Fig. 1a) as described previously[7,32]. The mice displayed anxiety-like behaviors, as demonstrated by the significantly reduced time and travel distance in the central region in the open field (OF) test, and the shorter time and fewer entries to the open arms in the elevated plus maze (EPM) test (Supplementary Fig. 1b, c). In addition, the CRS mice exhibited a greater extent of dextran sodium sulfate (DSS)-induced colitis, as evaluated by the loss of body weight, gross bleeding, and shortening of colon length, as well as epithelial damage and lymphocyte infiltration of the distal colon revealed in the histological analysis (Supplementary Fig. 1d–g). Consistently, the mRNA levels of proinflammatory cytokines (interleukin (IL) 6 (Il6), Il1b, Il12, and transforming growth factor beta (Tgfb))[33] were significantly increased in the colon tissues of CRS mice (Supplementary Fig. 1h). These results suggest that CRS could induce increase of anxiety-related behaviors and colitis susceptibility.

**Activation of AgRP neurons reverses CRS-increased anxiety-like behaviors and colitis susceptibility.** To investigate the involvement of AgRP neurons in CRS-induced effects, we conducted immunofluorescence (IF) staining for c-Fos, a signal reflecting neuronal activity[17], in the AgRP-Cre-Ai9 mice. IF staining of tdTomato (reflecting AgRP neurons) and c-Fos revealed decreased c-Fos levels in the AgRP neurons of CRS mice (Supplementary Fig. 2a), suggesting decreased AgRP neuronal activity. We then tested the effect of chemogenetic activation of AgRP neurons on CRS-increased susceptibility to anxiety and colitis using an excitatory DREADD receptor hM3Dq, which are activated by the inert ligand clozapine N-oxide (CNO)[34]. As predicted, stimulation of AgRP neurons (as shown by the increased c-Fos staining and food intake, Supplementary Fig. 2b–d) reversed CRS-induced anxiety-like behaviors with an increase in both center time duration and center distance in the OF test (Fig. 1a), and an increase in time and entries into the open arms in the EPM test (Fig. 1b). Furthermore, the activation of AgRP neurons also reversed chronic stress-increased susceptibility to DSS-induced colitis, as demonstrated by its blocking effects on the CRS-induced loss of body weight, increased bleeding score, shortened colon length, higher histological scores, and increased expression of proinflammatory factors (Il6, Il1b, Il12, and Tgfb) (Fig. 1c–g, and Supplementary Fig. 2e). Although activation of AgRP neurons has a significant impact on feeding behavior[17], the effect of colitis-related findings was not due to food intake as shown by pair-fed experiments (Supplementary Fig. 3a–e). It is worth mentioning that activation of AgRP neurons influence water intake[35], however, the signs of colitis were not worsened by increased DSS intake (Supplementary Fig. 2f). These results suggest that activation of AgRP neurons could reverse anxiety-related behaviors and colitis susceptibility induced by CRS.

**Inhibition of AgRP neurons promotes anxiety-like behaviors and colitis susceptibility.** To further confirm the role of AgRP neurons in anxiety and colitis, we investigated the phenotypes in mice with inhibition of AgRP neurons, using an inhibitory hM4Di designer receptor exclusively activated by designer drugs (DREADDs)[17], as reflected by the reduced c-Fos immunoreactivity in AgRP neurons (Supplementary Fig. 4a, b). Interestingly, inhibition of AgRP neurons decreased the center distance and center time in the OF test, and the number of entries and time spent in the open arms in the EPM test, indicating increased anxiety-like behaviors (Fig. 2a, b). Moreover, mice with inhibited neuronal activity of the AgRP were more sensitive to DSS-induced colitis, characterized by a more severe weight loss, gross bleeding, shortened colon length, higher histological scores, and increased pro-inflammatory cytokine levels (Il6, Il1b, Il12, and

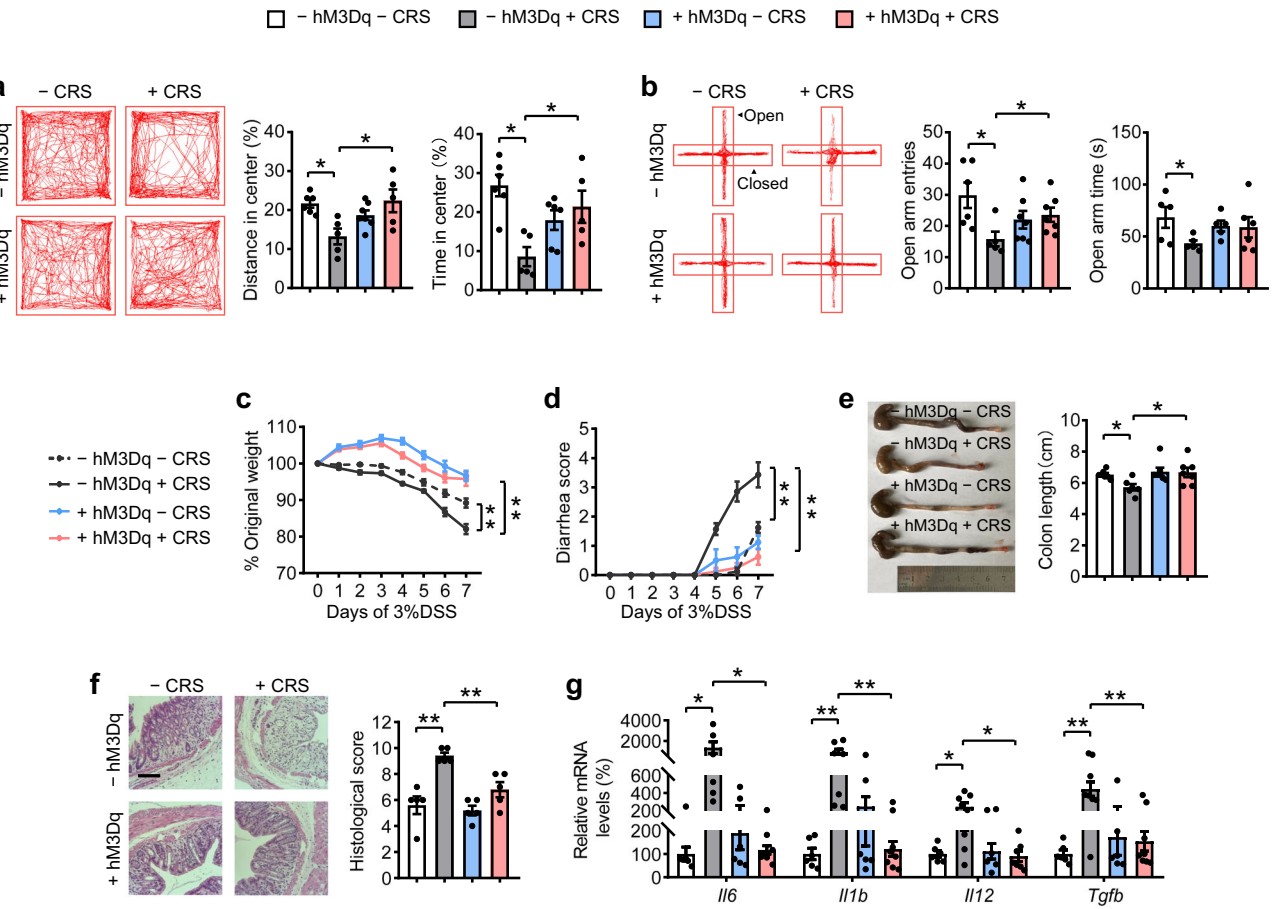

**Fig. 1 Activation of AgRP neurons reverses CRS-induced anxiety behaviors and colitis. a** Representative tracks and statistical results in OF test. **b** Representative tracks and statistics in EPM test. **c** Percentage of body weight loss. **d** Scores of diarrhea. **e** Gross morphology and length of the colon. **f** H&E staining and histological scores of the colon tissues. Scale bar, 110 μm. **g** qRT-PCR analysis of mRNA expression of inflammatory cytokines (*Il6, Il1b, Il12,* and *Tgfb*) in the distal colon tissues. Studies for **a**, **b** were conducted using 10–12-week-old *AgRP-Cre* mice receiving AAV expressing mCherry (− hM3Dq) or hM3Dq (+ hM3Dq), all mice experienced unstressed (− CRS) or stressed (+ CRS) treatment for 14 days. Behavioral tests were performed 30 min after single CNO injection on day 15 (**a**) and day 16 (**b**). **c–g** were performed using − hM3Dq mice and + hM3Dq mice under treatment of 3% DSS in drinking water for 7 days to induce acute colitis with (+ CRS) or without (− CRS) stress, simultaneously receiving CNO injections every 12 h per day. Values are expressed as means ± SEM (*n* = 5–8 per group), with individual data points. Data were analyzed using two-way analysis of variance, followed by Tukey's multiple comparisons test. *$P < 0.05$, **$P < 0.01$.

*Tgfb*), compared with control mice (Fig. 2c–g and Supplementary Fig. 4c). Collectively, these results indicate that inhibition of AgRP neurons mimics stress-induced anxiety-like behaviors and colitis susceptibility.

**Overexpression of c-Jun in AgRP neurons confers resistance to CRS-induced anxiety-like behaviors and colitis susceptibility.** We then explored the possible involvement of c-Jun in the CRS-induced effects and found that the activity of AgRP neurons decreased through c-Jun ablation and increased through c-Jun overexpression both in vivo and in vitro (Supplementary Fig. 5a–d). IF staining confirmed a decrease in c-Jun protein levels in the AgRP neurons of stressed mice (Supplementary Fig. 6a). If the reduced c-Jun expression was important under stress, the activation of c-Jun in AgRP neurons should be expected to ameliorate CRS-induced effects. To test it, we overexpressed *c-Jun* in AgRP neurons by injecting AAVs expressing c-Jun into the *AgRP-irs-Cre* mice (Supplementary Fig. 6b, c). Consistently, overexpressed groups spent more time and distance in the center as evaluated in the OF test compared with the control group after CRS (Supplementary Fig. 6d). Similarly, in the EPM test, groups with overexpressed c-Jun had more entries to

the open arm (Supplementary Fig. 6e). Stressed c-Jun-overexpressing mice were more resistant to DSS-induced body weight loss compared with stressed control mice (Supplementary Fig. 6f, g). The diarrhea scores and colon length were also relieved in the overexpressed group (Supplementary Fig. 6h, i). Furthermore, signs of colon colitis were markedly ameliorated in mice with overexpressed c-Jun, as evidenced by the decreased epithelial damage and lymphocyte infiltration, as well as reduced mRNA expression of inflammatory cytokines (Supplementary Fig. 6j, k). These data indicate that overexpression of c-Jun in AgRP neurons protects the mice from CRS-induced anxiety and colitis susceptibility.

**Deletion of c-Jun in AgRP neurons facilitates anxiety-like behaviors and colitis.** To further confirm the role of c-Jun in AgRP neurons, we generated mice with *c-Jun* knockdown in AgRP neurons ($c\text{-}Jun^{\Delta AgRP}$), as confirmed by the reduced c-Jun expression in AgRP neurons (Supplementary Fig. 7a). The corticosterone concentration was significantly higher in the serum of $c\text{-}Jun^{\Delta AgRP}$ mice than in control mice, suggesting increased stress (Supplementary Fig. 7b). Moreover, the body weight of $c\text{-}Jun^{\Delta AgRP}$ mice was slightly lower than that of control mice (Supplementary Fig. 7c).

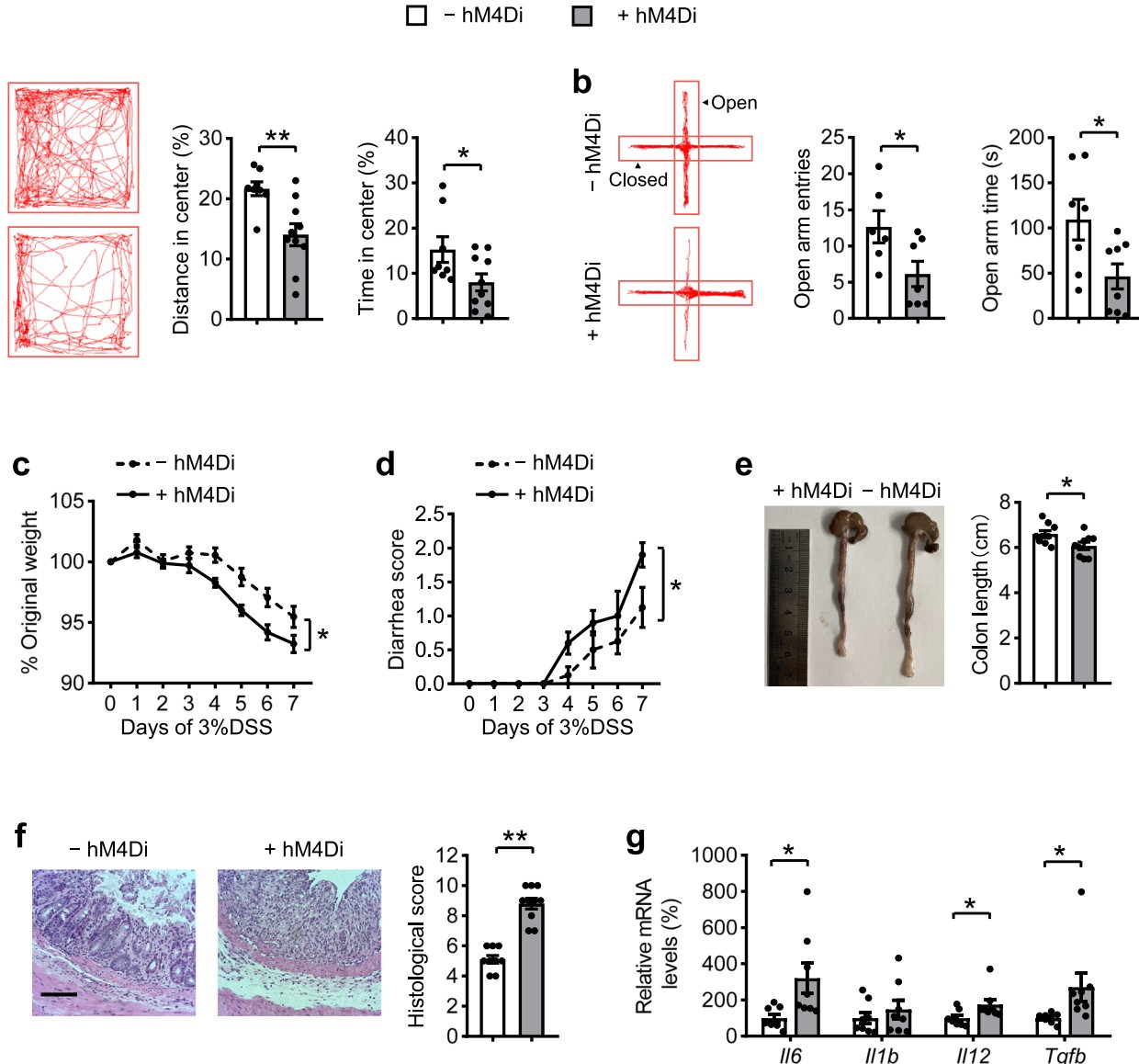

**Fig. 2 Inhibition of AgRP neurons mimics CRS-increased susceptibility to anxiety and colitis. a** Representative tracks and statistical results in OF test. **b** Representative tracks and statistics in EPM test. **c** Percentage of body weight loss. **d** Scores of diarrhea. **e** Gross morphology and length of the colon. **f** H&E staining and histological scores of the colon tissues. Scale bar, 110 μm. **g** qRT-PCR analysis of mRNA expression of inflammatory cytokines (*Il6, Il1b, Il12,* and *Tgfb*) in the distal colon tissues. Studies for **a**, **b** were conducted using 10–12-week-old *AgRP-Cre* mice receiving AAV expressing mCherry (- hM4Di) or hM4Di (+ hM4Di), all mice received CNO injections every 12 h per day. Behavioral tests were performed 30 min after single CNO injection on day 22 (**a**) and day 23 (**b**). **c**–**g** were performed using − hM4Di mice and + hM4Di mice with 3% DSS in drinking water for 7 days to induce acute colitis after 21 days of CNO injections. Values are expressed as means ± SEM (*n* = 6–10 per group), with individual data points. Data were analyzed using two-way analysis of variance with Bonferroni's multiple comparisons test (**c**, **d**). Data were analyzed using two-tailed unpaired Student's *t*-test (**a**, **b**, **e**–**g**). *$P < 0.05$, **$P < 0.01$.

The c-Jun^ΔAgRP mice displayed obvious anxiety-like behaviors, reflected by a shorter time and less distance in the center in the OF test (Fig. 3a), and fewer entries and time in the open arms as evaluated in the EPM test (Fig. 3b). The clinical signs of colitis, including weight loss, rectal bleeding and colon shortening, were more severe in c-Jun^ΔAgRP mice than in controls after DSS treatment (Fig. 3c–e). The epithelial damage, including mucosal erosion, crypt loss, lymphocyte infiltration, and the mRNA expression of proinflammatory cytokines were also significantly increased in the colons of c-Jun^ΔAgRP mice compared with controls (Fig. 3f, g). These data indicate that deletion of c-Jun in AgRP neurons is sufficient to induce anxiety-like behaviors and colitis susceptibility in the absence of stress.

**The increased colitis susceptibility in c-Jun^ΔAgRP mice is mediated by THBS1.** Because the brain conveys the neural, endocrine, and circulatory messages to the gut[8–10]. To elucidate the underlying mechanisms of the observed effects, we conducted mass spectrometry to explore the possible secreted proteins in the serum of the mice with c-Jun overexpression and control groups with or without CRS under colitis conditions. We identified 22 secreted proteins that significantly differed in abundance between the three groups (Fig. 4a, b and Supplementary Tables 1, 2). Among these proteins, we focused on thrombospondin 1 (THBS1), which showed the most dramatic change and is well known for its anti-angiogenic and anti-inflammatory properties[36,37]. The secreted levels of THBS1 were significantly

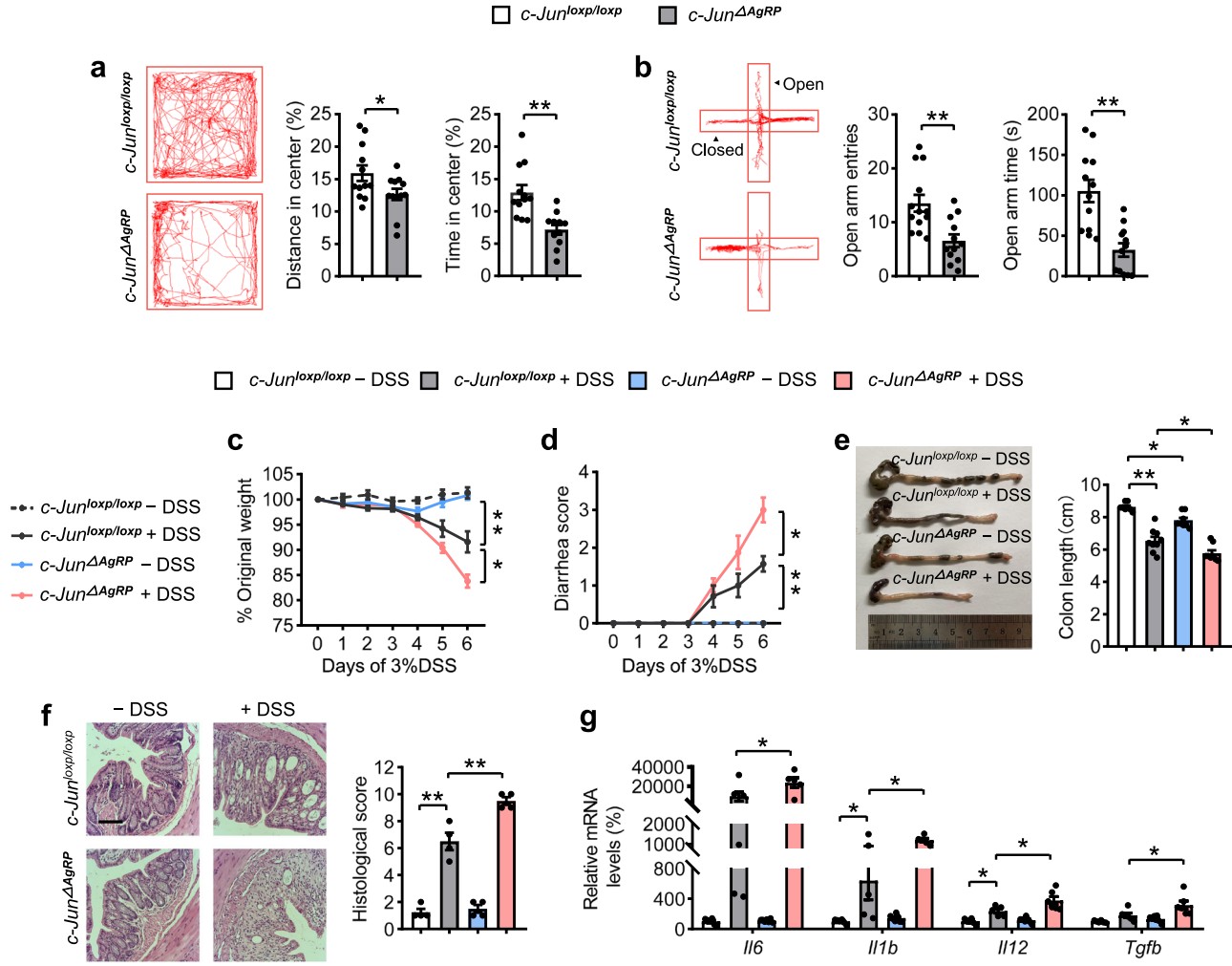

**Fig. 3 Deletion of c-Jun in AgRP neurons facilitates anxiety-like behaviors and colitis. a** Representative tracks and statistical results in OF test. **b** Representative tracks and statistics in EPM test. **c** Percentage of body weight loss. **d** Scores of diarrhea. **e** Gross morphology and length of the colon. **f** H&E staining and histological scores of the colon tissues. Scale bar, 110 μm. **g** qRT-PCR analysis of mRNA expression of inflammatory cytokines (*Il6, Il1b, Il12,* and *Tgfb*) in the distal colonic tissues. Studies for **a**, **b** were conducted using 20–22-week-old control mice (*c-Jun*^loxp/loxp) or mice with *c-Jun* deletion in AgRP neurons (*c-Jun*^ΔAgRP). **c**–**g** were performed in *c-Jun*^loxp/loxp and *c-Jun*^ΔAgRP mice administrated with (+ DSS) or without (− DSS) 3% DSS for 6 days to induce acute colitis. Values are expressed as means ± SEM (*n* = 4–12 per group), with individual data points. Data were analyzed using two-tailed unpaired Student's *t*-test (**a**, **b**). Data were analyzed using two-way analysis of variance, followed by Tukey's multiple comparisons test (**c**–**g**). \**P* < 0.05, \*\**P* < 0.01.

reduced after CRS and notably increased after c-Jun rescue in AgRP neurons (Supplementary Fig. 8a–c). Moreover, the serum levels of THBS1 were reduced in *c-Jun*^ΔAgRP mice and DSS treatment mice (Fig. 4c and Supplementary Fig. 8d) indicating it may function downstream of c-Jun. To test this hypothesis, we treated *c-Jun*^ΔAgRP mice with THBS1[38] and found that it markedly alleviated colitis, as shown by the resistant effects on the corresponding changes in the body weight loss, bleeding score, colon length, colon histochemical analysis, and the expression of pro-inflammatory factors (Fig. 4d–h). These results suggest that THBS1 suppresses intestinal mucosal inflammation and may serve as a potential biomarker for stress-induced colitis susceptibility.

## Discussion

The brain-gut axis serves as a circuit that incorporates the state of mind and gut signals that ultimately determine the intestinal function[8–10]. Therefore, the changes of brain functions are closely related to gut metabolism abnormalities[8,10,39]. Accumulating evidence indicates that mood disorders, such as anxiety or depression, often co-occur with IBD[4,40–42]. Several brain areas, including the hypothalamus, hippocampus and amygdala, are involved in anxiety-related behaviors[11,32,43]. The AgRP neurons in the hypothalamic ARC particularly have gained much attention since more important functions of these neurons are discovered, such as feeding, pain sensation and depression-related behaviors[14,15,18]. However, the role of AgRP neurons in the comorbidity of anxiety and colitis has not been reported.

In the current study, we used CRS model and chemogenic strategy to investigate the possible involvement of AgRP neurons in stress-induced anxiety and colitis susceptibility. We found that AgRP neuronal activity was inhibited by CRS. The significance of the inhibited AgRP neurons in CRS was further confirmed by gain- and loss-of AgRP neuron function experiments. Although some recent studies have shown that AgRP neurons are involved in chronic unpredictable stress mediated depression-related behaviors and engaged in high-fat diet-induced anxiety and depression[3,14]. However, less studies focus on the co-occurrence of anxiety and gut inflammation at the animal levels. This study is important to demonstrate that convergent regulation of colitis

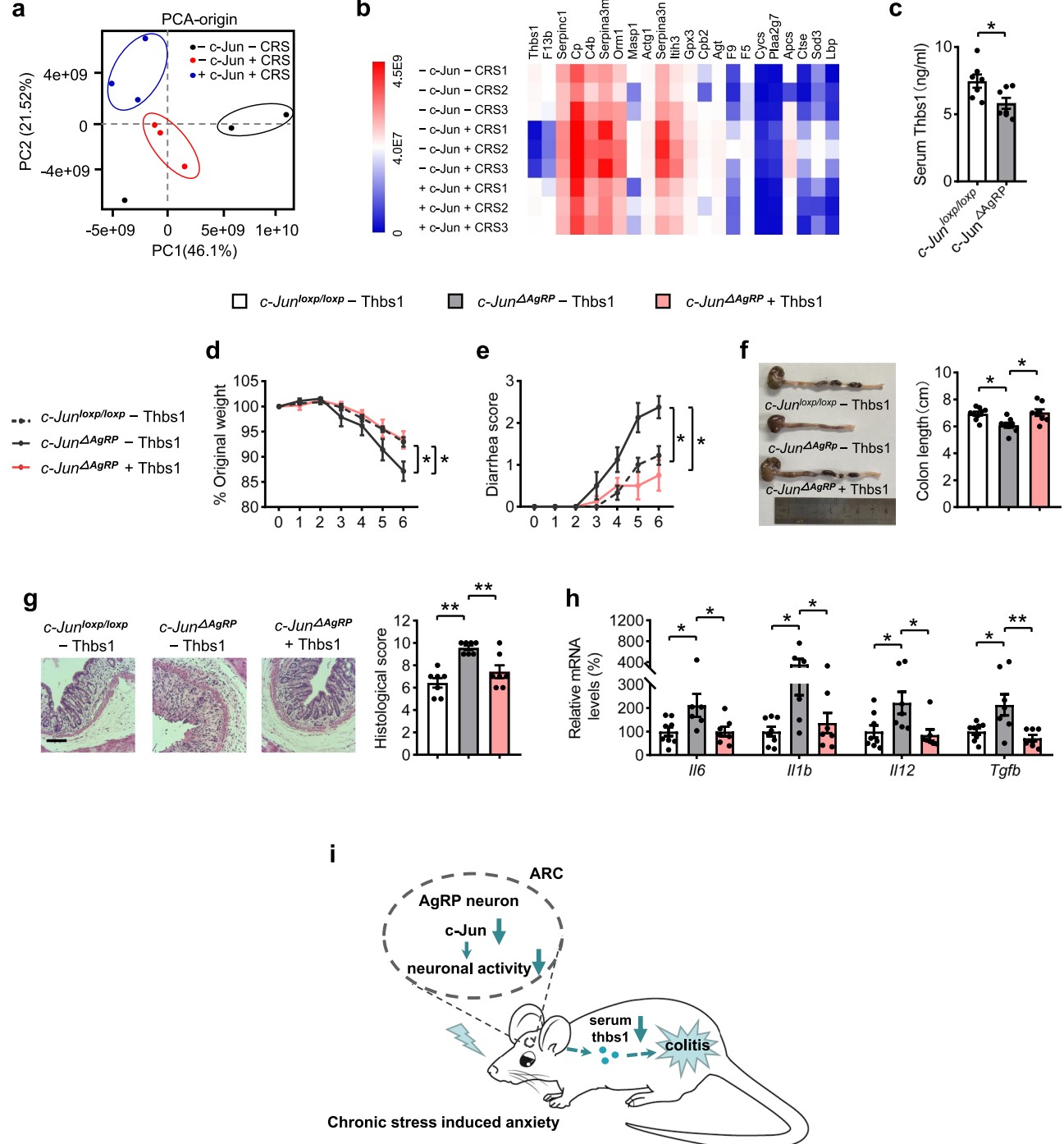

**Fig. 4 The increased colitis susceptibility in *c-Jun*$^{\Delta AgRP}$ mice is mediated by THBS1. a** Partial least-squares discriminant analysis of protein composition. **b** Spearman's correlation analysis of serum proteome. **c** Serum THBS1 levels. **d** Percentage of body weight loss. **e** Scores of diarrhea. **f** Gross morphology and length of the colon. **g** H&E staining and histological scores of the colon tissues. Scale bar, 110 μm. **h** qRT-PCR analysis of mRNA expression of inflammatory cytokines (*Il6, Il1b, Il12,* and *Tgfb*) in the distal colonic tissues. **i** Summary Diagram. Chronic restraint stress induces anxiety-like behaviors and colitis susceptibility, which is mediated by c-Jun in AgRP neurons. Knockdown of *c-Jun* in AgRP neurons increases colitis susceptibility through reducing serum THBS1 levels. Studies for **a**, **b** were conducted using - c-Jun - CRS mice, - c-Jun + CRS mice and + c-Jun + CRS mice with 3% DSS for 7 days. Serum was collected after DSS stimulation for proteomics profiling. **c** was conducted using 24–26-week-old *c-Jun*$^{loxp/loxp}$ mice and *c-Jun*$^{\Delta AgRP}$ mice with DSS administration. **d**–**h** were conducted using 22–24-week-old *c-Jun*$^{loxp/loxp}$ mice and *c-Jun*$^{\Delta AgRP}$ mice with (+ thbs1) or without (- thbs1) THBS1 supplementary, simultaneously receiving 3% DSS treatment for 6 days to induce acute colitis. Values are expressed as means ± SEM (*n* = 3–9 per group), with individual data points. Data were analyzed using two-tailed unpaired Student's *t*-test (**c**). Data were analyzed using two-way analysis of variance, followed by Tukey's multiple comparisons test (**d**, **e**). Data were analyzed using one-way analysis of variance, followed by Tukey's multiple comparisons test (**f**–**h**). *$P < 0.05$, **$P < 0.01$.

and stress is integrated in the AgRP neurons in the ARC, providing important evidence that psychiatric disorders, such as anxiety, may influence colitis through central neuronal activity in the ARC. Moreover, though some mechanisms are proposed for IBD[44], the neuronal signals are largely unknown. Our results provide new perspective for understanding the neuronal regulation for IBD. It is worth mentioning that AgRP neurons were involved in stress induced colitis susceptibility rather than spontaneous colitis, implying that the pathology of colitis required conditions of DSS stimulation. We used a 14-day restraint model and followed DSS administration to certify that AgRP neurons were important in the stress-induced co-occurrence of anxiety and colitis susceptibility. However, this does not imply that changes in AgRP neurons always accompany these two diseases. As several studies have shown that AgRP neurons facilitate food-seeking behavior[17,19], it remains to be tested whether the role of AgRP neurons in anxiety-like behaviors is influenced by the behaviors driven by feeding. However, the increased food intake did not contribute to the improvement of colitis as evaluated by pair-fed experiments in the current study.

We then explored the specific neural molecules regulating stress-induced anxiety and colitis. We found that expression of the stress-related gene c-Jun was decreased in AgRP neurons under CRS conditions. Furthermore, knockdown of *c-Jun* in AgRP neurons mimicked the effect of CRS but reversed these after overexpression of *c-Jun*. These results demonstrate a regulatory role of the central c-Jun in stress-induced anxiety and colitis susceptibility and provide a potential therapeutic approach for the treatment of these diseases. However, it is unclear how c-Jun inhibited neuronal activity under stress. Considering that c-Jun plays a role in the regulation of neuronal injury and axon regeneration[24,25], we speculated that stress may lead to the damage of AgRP neurons, thus leading to a change in neuronal activity. However, this possibility requires further investigation.

To gain further insights into how anxiety influences colitis, we performed proteomic analysis and serum measurement and found that the levels of secreted protein THBS1 were decreased by CRS in a c-Jun dependent manner, suggesting that it may function as downstream in linking c-Jun regulated colitis. Thrombospondins are a family of extracellular matrix proteins, which are first identified in platelets stimulated with thrombin[45]. After treatment with DSS, THBS1-deficient mice show a higher level of crypt damage and deeper lesions, which are reversed by treatment with a THBS1 mimetic peptide[46]. Besides, another THBS1 mimicking peptide, ABT-898, has been shown to alleviate DSS-induced colitis in WT mice[47]. Moreover, the 3TSR treatment, which includes three type 1 repeats of the THBS1, is effective in attenuating the inflammatory response to DSS injury[48]. The importance of THBS1 in mediating anxiety-associated colitis was confirmed by the reversal effect of recombinant THBS1 on colitis in *c-Jun*$^{\Delta AgRP}$ mice. We also found that supplementary of THBS1 had no relief effect on anxiety-like behaviors in *c-Jun*$^{\Delta AgRP}$ mice (Supplementary Fig. 8e, f). These results indicate that THBS1 might act downstream of c-Jun in regulation of gut inflammation rather than acting on the brain. Because THBS1 levels were correspondingly changed with the status of anxiety and colitis, suggesting that it might be used as a biomarker for the comorbidity of these diseases. However, it remains unclear for the source of secreted THSB1 protein in the serum, as THBS1 can be produced and secreted into the extracellular space of many cell types, including the activated endothelium, intestinal epithelial cells, and astrocytes[36,49,50]. Studies have shown that THBS1 can be secreted by astrocytes and promotes CNS synaptogenesis[49]. Consistently, other studies have revealed that neuron-to-astrocyte communication is established when the physiological state changes. The activation of AgRP neurons can promote changes in

neighboring astrocytes by releasing the inhibitory neurotransmitter GABA[51]. Therefore, we speculated that altered AgRP neuronal activity may stimulate the neighboring astrocytes, which in turn causes the secretion of THBS1. This issue remains to be determined. In addition, several lines of evidence illustrate that the vagus nerve provides parasympathetic innervation to the gastrointestinal tract, coordinating the interactions between central and peripheral, which exerts influence on the development of inflammatory bowel disease[52,53]. Therefore, changes in central nervous system may affect the gut through the vagus nerve, since AgRP neurons show a close connection with vagal[54]. Besides, intestinal epithelial cells can produce THBS1 and regulate intestinal inflammatory responses through modulating monocytes properties[50]. These studies indicate the possibility of a connection between the neuron and the secretory protein. These questions remain for future investigation.

Interestingly, we found that chemogenic manipulation or *c-Jun* knockdown-induced inhibition of AgRP neurons promotes anxiety-like behaviors and colitis susceptibility in the absence of stressor, our results are important for understanding the mechanisms for those without obvious stress but present with the comorbidity of psychiatric disorders and IBD. Previous studies have also described sexual dimorphism in the regulation of psychiatric disorders and inflammatory bowel disease[55,56]. The sex hormones might account for the pathogenesis of the two diseases[57,58]. The underlying mechanisms need to be further investigated.

In summary, our present findings revealed that AgRP neuronal activity in the ARC is an important link between anxiety-like behavior and intestinal inflammation (Fig. 4i). The importance of these findings is that we have uncovered the specific neurons and signals in the brain underlying the regulation of the anxiety and colitis comorbidity. Our results provide evidence that CRS-induced anxiety and colitis susceptibility is mediated through an unexpected neurons AgRP neurons. Moreover, we demonstrated c-Jun as a target in AgRP neuron for stress-induced anxiety and colitis susceptibility. Furthermore, we identified the secreted protein THBS1 function in linking anxiety and colitis and as a biomarker for anxiety-colitis comorbidity. These results provide a new perspective for exploring the brain in the regulation of intestinal inflammation homeostasis, and further provide a new central target for the therapeutic intervention of stress-induced psychiatric disorders and intestinal metabolism dysfunction.

## Methods

**Mice and treatment**. Adult male C57BL/6 wild-type (WT) mice were purchased from Shanghai Laboratory Animal Co., Ltd. (Shanghai, China). *c-Jun*$^{loxp/loxp}$ mice (generously provided by Dr. Erwin F. Wagner, Cancer Cell Biology Program, Spanish National Cancer Research Center) were crossed with mice expressing Cre recombinase under control of the AgRP promoter (*AgRP-irs-Cre* mice) to generate *c-Jun*$^{\Delta AgRP}$ mice. To characterize the changes of signaling in AgRP neurons, we used the Ai9 (tdTomato) reporter system, which is Cre-driver lines for cell-type-specific genetic manipulation[59]. The efficiency of AgRP-specific *c-Jun* deletion was evaluated by mating Ai9 reporter mice with *c-Jun*$^{\Delta AgRP}$ transgenic mice or *AgRP-irs-Cre* mice. Ai9 (tdTomato) reporter mice and *AgRP-irs-Cre* mice were both obtained from Jackson Laboratory (Bar Harbor, ME, USA). Transgenic mice and their littermates were used in experiments at the indicated ages. Mice were subcutaneously bilaterally injected with recombinant human thrombospondin 1 (THBS1) protein (0.5 mg/kg per day; Novoprotein Scientific Inc., Shanghai, China)[38] or vehicle (phosphate-buffered saline) every day for 6 days. Food intake was measured in the light cycle based on the feed condition.

Mice were maintained under controlled temperature (23℃), humidity (50–60%), and illumination (12-h light/12-h dark cycle), and provided *ad libitum* access to food and water. All animals used in this study were male. All animal experiments were conducted in accordance with the guidelines of the Institutional Animal Care and Use Committee of Shanghai Institute for Nutritional Sciences, Chinese Academy of Sciences (SINH-2022-GFF-1).

**Chronic Restraint Stress (CRS) model**. The CRS mouse model was performed as described previously[7,32,60]. In brief, the mice were individually placed in a 50-mL

polypropylene conical tube with multiple holes for ventilation and were restrained to prevent back-and-forth movement. The mice were not available to food and water during the stress. Once the stress ended, mice were put back to their home cages immediately with access to food and water freely. For the control group, the mice were placed in the home cage at the same time without food and water. Restraint was applied for 4 h per day from 10:00 a.m. to 2:00 p.m. for the number of days indicated.

**Colitis model establishment**. To establish the dextran sulfate sodium (DSS)-induced colitis model, the drinking water of the mice was supplemented with 3% (w/v) DSS (40 kDa; Aladdin, Shanghai, China) as described previously[33], and the colon length was determined at the end of the experiments. Diarrhea scores were assessed as described previously[61].

**Stereotaxic surgery and viral injections**. Surgery was performed as reported previously[62] with a stereotaxic frame (Stoelting, Wood Dale, IL, USA). Viral injection coordinates (in mm, midline, bregma, dorsal surface) are as follows: for ARC (±0.3, −1.5, −5.9)[3,17]. To rescue the expression of *c-Jun* specifically localized in AgRP neurons, *AgRP-irs-Cre* mice were bilaterally injected with 300 nL of a Cre-dependent adeno-associated virus (AAV) vector containing *c-Jun* in the opposite orientation flanked by two inverted loxP sites [AAV9-EF1a-DIO-c-Jun-mCherry, $2.5 \times 10^{12}$ particle-forming units (PFU)/mL] into the ARC, or with an AAV vector containing only mCherry in the opposite orientation flanked by two inverted loxP sites (AAV9-EF1a-DIO-mCherry, $2.5 \times 10^{12}$ PFU/mL) as controls.

**Designer Receptor Exclusively Activated by Designer Drugs (DREADDs)**. To inhibit AgRP neuronal activity, *AgRP-irs-Cre* mice were stereotaxically injected with 300 nL of a Cre-dependent AAV encoding an inhibitory DREADD GPCR (hM4Di) (AAV9-EF1a-DIO-hM4Di-mCherry, $8 \times 10^{12}$ PFU/mL) or an AAV encoding only mCherry (AAV9-EF1a-DIO-mCherry, $7 \times 10^{12}$ PFU/mL) as controls, bilaterally into the ARC.

For chemogenetic activation of AgRP neurons, *AgRP-irs-Cre* mice were stereotaxically injected with 300 nL of a Cre-dependent AAV encoding an excitatory DREADD GPCR (hM3Dq) (AAV9-EF1a-DIO-hM3D(Gq)-mCherry, $3 \times 10^{12}$ PFU/mL) or an AAV encoding only mCherry (AAV9-EF1a-DIO-mCherry, $3 \times 10^{12}$ PFU/mL) as controls, bilaterally into the ARC. After 3 weeks of recovery, all mice were then intraperitoneally injected with clozapine N-oxide (CNO) (MedChemExpress, NJ, USA) at 0.3 mg/kg of body weight for hM4Di[17] silencing and for hM3Dq activation[34] every 12 h for indicated days. CNO was administered 30 min before the behavioral test.

**Paired feeding assays**. Pair-feeding (PF) experiments were performed as previously described[63]. In brief, mice were receiving AAV expressing mCherry (−hM3Dq) or hM3Dq (+ hM3Dq) and individually housed. Food intake was recorded during 14 days of stress. Subsequently, the − hM3Dq and + hM3Dq mice were randomly paired. The PF experiment was conducted during CNO injection. The PF mouse was given an equivalent amount of food consumed the previous day by its paired partner.

**Isolation and treatment of primary hypothalamic neurons**. The mouse primary cultures of hypothalamic neurons were referred as previously described[64]. sh-*c-Jun* and *c-Jun* over-expressed (pCMV-*c-Jun*) plasmid were transfected into cells using lipofectamine 3000 reagent (Invitrogen; Carlsbad, CA, USA) according to the manufacturer's recommendation. The shRNA sequence for mouse *c-Jun* was 5'-GGCACAGCTTAAGCAGAAAGT-3'.

**Open Field (OF) test**. The OF test was performed as in previous studies[32,60]. In brief, a white open field box (50 × 50 × 50 cm; length × width × height) was divided into a center field (25 × 25 cm) and a periphery field for analysis purposes. The track was analyzed using LabState (AniLab) by recognizing the central body point of the mouse throughout a 10-min session. Less time and locomotion spent in the center of the box were interpreted as anxiety-like behaviors.

**Elevated Plus Maze (EPM) test**. The EPM test was performed as previously described[32,60]. The elevated plus maze made of plastic and consisted of two white open arms without walls and two white enclosed arms with walls (25-cm long, 5-cm wide, 15-cm high). The maze was placed 60 cm above the floor. Mice were introduced into the center quadrant with their back facing an open arm. The ANY-maze video tracking system (Anilab) was used to track and analyze the time of mice spent in the open arms and their entries into the open arms throughout a 10-min session. Anxiety was evaluated by fewer movements into the open arms and less time spent there.

**RNA Isolation and Quantitative Real-time (qRT)-PCR**. RNA extraction and qRT-PCR were performed as described previously[64]. The primer sequences used in this study are provided in Supplementary Table 3.

**Histological scoring**. Histological scoring was performed as described previously[65]. Hematoxylin and eosin (H&E)-staining of colonic tissue sections were scored in a blinded fashion for determining the degree of inflammation and tissue damage. Colonic epithelial damage was scored as follows: 0 = normal; 1 = hyperplasia, irregular crypts, and goblet cell loss; 2 = mild to moderate crypt loss (10–50%); 3 = severe crypt loss (51–90%); 4 = complete crypt loss, surface epithelium intact; 5 = small to medium sized ulcer (<10 crypt widths); 6 = large ulcer (>10 crypt widths). Infiltration was scored separately for mucosa (0 = normal; 1 = mild; 2 = modest; 3 = severe), submucosa (0 = normal; 1 = mild to modest; 2 = severe), and muscle/serosa (0 = normal; 1 = moderate to severe). Scores for epithelial damage and inflammatory cell infiltration were added, resulting in a total scoring range of 0–12.

**Serum measurements**. The proteomics was performed by the serum after removing common high-abundance protein through using a Thermo Fisher's serum High Abundance protein removal reagent (High Select™ Top14 Abundant Protein Depletion Mini Spin Columns, A36370, Thermo fisher, US). The mass spectrometer Thermo Scientific Q Exactive was performed for Label-free quantification detection and data were analyzed by Proteome Discoverer 2.2. The secreted proteins with fold change (FC) ≥ 1.5 and *p*-value < 0.05 were considered to be differential proteins. Volcano plots were used to filter the proteins of interest which based on $\log_2$(FC) and $-\log_{10}$(P-value) of the secreted proteins[66]. The raw data is available in the link: https://doi.org/10.6084/m9.figshare.21776357.v1. Serum corticosterone levels were measured with an enzyme-linked immunosorbent assay (ELISA) kit (ADI-900-097; Enzo Life Science, Farmingdale, NY, USA) as described previously[67]. The THBS1 levels were measured using an ELISA kit (mlbio, Shanghai, China) according to the manufacturer's recommendations.

**Immunofluorescence (IF) staining**. Immunofluorescence staining was performed as described previously[62] with primary antibodies to c-Jun (1:1000, Cell Signaling Technology, Danvers, MA, USA) and c-Fos (1:1000, Cell Signaling Technology or 1:500, Santa Cruz Biotechnology, Santa Cruz, CA, USA). c-Fos staining was coupled with a TSA Plus Fluorescein KIT (NEL741001KT, Perkinelmer, Waltham, MA, USA).

**Statistics and reproducibility**. Experimental data are expressed as the mean ± standard error of the mean (SEM) of the number of tests stated for each experiment. Unpaired Student's *t*-test was used for two-group comparisons. Data with two factors were analyzed by two-way analysis of variance (ANOVA) with Bonferroni's multiple comparison. For multiple group comparisons, ordinary one-way ANOVA or two-way ANOVA with Tukey's multiple comparisons test was used. All statistical tests were performed using GraphPad Prism, version 8.0 (GraphPad Software, San Diego, CA, USA). In addition, the individual data points on each graph were shown in order to reflect the individual variability of the measures. *$P < 0.05$, **$P < 0.01$.

**Reporting summary**. Further information on research design is available in the Nature Research Reporting Summary linked to this article.

## Data availability
The source data underlying the graphs and charts in the main manuscript file are shown in Supplementary Data 1. The raw data of serum proteomics is available in the link: https://doi.org/10.6084/m9.figshare.21776357.v1. Other data that support the study are available from the corresponding author upon reasonable request.

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

## Acknowledgements

We would like to thank Dr. Erwin F. Wagner (Cancer Cell Biology Program, Spanish National Cancer Research Center) and Lijian Hui (Chinese Academy of Sciences Center for Excellence in Molecular Cell Science) for providing c-Jun loxp/loxp mice.

This work was supported by grant from the National Natural Science Foundation (31830044, 91957207, 81870592, 82170868, 81770852, 81970742, 82270905, 81970731, 82000764 and 91942312), The National Key R&D Program of China (2018YFA0800600), and Shanghai leading talent program, CAS Interdisciplinary Innovation Team, Novo Nordisk-Chinese Academy of Sciences Research Fund (NNCAS-2008-10). Natural Science Foundation of Shanghai "science and technology innovation action plan" (21ZR1475900).

## Author contributions

Fuxin Jiao and Feifan Guo designed the project and analyzed the data; Fuxin Jiao, Xiaoming Hu, Hanrui Yin, Feixiang Yuan, Ziheng Zhou performed the experiments; Fuxin Jiao and Feifan Guo wrote the manuscript. Zhanju Liu edited the manuscript, provided technical support, contributed to research design and discussion. Wei Wu and Shanghai Chen provided experimental materials; All authors discussed and revised the manuscript.

## Competing interests

The authors declare no competing interests.
