## [Peer Review File · Communications Biology]

Reviewers' comments:

Reviewer #1 (Remarks to the Author):

This paper described interesting observations on AgRP neuron action in control anxiety-related behaviors and the susceptibility to colitis. Multiple approaches including chemogenic manipulations of AgRP neurons as well animal models with c-Jun specific AgRP KO were used to test whether AgRP neuron activity is sufficient and required for the susceptibility to colitis. A series of strong datasets support that AgRP neurons are inhibited by stress, which in turn increases the susceptibility to colitis induced by DSS. Although the results on anxiety-like were not novel, the impact of colitis seems to be completely novel. The experiments were appropriately designed and analyzed. However, some issues were noted during the review.

1) While it is known the intestinal lumen can undergo dynamic turnover, the effect on the colon length seems to be rather surprising as the results showed that over 1 week of treatments exhibited a dramatic change in the length up to 2cm. This set of data may be benefited from further considerations. In addition, the baseline length in controls appears to be very variable, e.g. Fig. 3E the control length was shown to be about 8.5cm while the control in other experiments were shown to be about 6.5cm. At the minimum, the authors should discuss about this including the potential underlying mechanism responsible for this dramatic change in length.

2) Since acute AgRP neuron activity changes are known to alter feeding and drinking, the potential different amount of DSS intake due to changes in drinking behavior on the observed outcome needs to be considered.

3) In Fig. 4, was the proteomic study done with or without the DSS treatment? What about the effect of DSS treatments alone on THBS1?

Reviewer #2 (Remarks to the Author):

This study investigated the roles of c-Jun in hypothalamic AgRP neurons in stress-induced anxiety and colitis susceptibility. They used chronic restraint stress (CRS) to generate increased anxiety and colitis susceptibility as a condition of psychiatric disorder and inflammatory bowel disease (IBD). Firstly, they compared CRS mice and control mice to show decreased AgRP neuronal activity in CRS group. That suggested these AgRP neurons were inhibited when CRS induced anxiety and colitis susceptibility. To further investigate the functions of AgRP neurons in stress-induced anxiety and colitis susceptibility, they used chemogenic strategy (DREADD-Dq) to specifically activate these AgRP neurons. The activation of AgRP neurons could rescue CRS-increased anxiety behaviors and colitis susceptibility. It suggested the inhibition of AgRP neurons is required to stress-induced anxiety and colitis susceptibility. Then they also used DREADD-Di to specifically inhibit these AgRP neurons. The inhibition of AgRP neurons could induce anxiety behaviors and colitis susceptibility. It suggested the inhibition of AgRP neurons is sufficient to stress-induced anxiety and colitis susceptibility. In addition, they explored the critical role of c-Jun in AgRP neurons in mediating stress-induced anxiety and colitis susceptibility. They confirmed the requirement and sufficiency of c-Jun in AgRP neurons via overexpression and deletion of c-Jun in the regulation of stress-induced anxiety and colitis susceptibility. Finally, they conducted mass spectrometry to screen candidates of secret proteins, and further validated thrombospondin 1 (THBS1) in colitis susceptibility. In summary, this study provides a new neuron-endocrine (c-Jun-AgRP-THBS1) to treat anxiety and IBD. As we know, psychiatric disorders are sex dimorphic. This study only included males, what about females? Did you try in females? Please clarify in discussion.

1 What is "pair-fed experiments"? You'd better to add the method.

2 How to treat THBS1? Method? Dose?

3 Main figures:

Figure 1B, bar graph, what about difference and significance of - hM3Dq + CRS and + hM3Dq + CRS?

Figure 4D-H, did you do c-Junloxp/loxp + Thbs1? If Thbs1 can improve colitis susceptibility in control group?

Figure 4, you only presented colitis susceptibility related. What about anxiety related: open field and elevated plus maze? Thbs1 involved in anxiety-like behavior or not?

4 Supplementary figures:

Supplementary F 2B, when received CNO? Please describe details.

Supplementary F 2D, food intake, dark cycle or light cycle? Feed or fasting? Please describe details.

Supplementary F 4, when inhibiting AgRP neurons, why the timeline of OFT, EPM and 3% DSS is not consistent to activation of AgRP and c-Jun?

Supplementary F 5, what about the expression of c-Jun? Please add the staining of c-Jun.

Supplementary F 7, you used AgRP-Cre-Ai9 as control in "A", but c-Junloxp/loxp as control in "B" and all other experiments about c-Jun. why not use c-Junloxp/loxp as control also in "A"?

5 The writing is not good and not easy reading.

1) There are too many long-complicated sentences which make it hard to read and understand. Please break down these long sentences.

2) Please make sentences direct and concise

Edits eg.

Page 3, line 52 and 53, there are three "of" in one sentence. Please make simple.

Page 3, line 60-64, "Unlike the POMC that food-seeking behavior" I can get your point, however the writing is not smooth. Please edit to make reading easily.

Page 4, line 68, could delete "either"; line 69, could change "and plays a role in the activation of" to "to activate".

Page 8, line 153, change "this possibility" to "it".

Page 9, line 174, like "sera", in this manuscript you sometimes use "sera", sometimes "serum". I suggest only "serum". That will be easy reading.

Line 356-376, please reorganize words.

Reviewer #3 (Remarks to the Author):

In this manuscript, Jiao et al systematically analyzed the neuronal and molecular mechanism underlying the co-occurrence of stress-induced increase in anxiety and colitis susceptibility. They found that CRS markedly decrease the activity of AgRP cells with a concomitant inhibition of c-Jun in this particular cell type. More importantly, artificial regulation of the AgRP cell activity or the c-jun expression was sufficient to mimic /antagonize the effect of CRS. Finally, they showed that the altered Thbs1 secretion may mediating some of the roles of AgRP cells on stress effect. Overall, the data presented herein help to expand our understanding of the neuronal underpinning of stress-induced occurrence of anxiety and colitis. However, there are also some important issues to be considered.

Major:

1. Although the authors have provided evidence to show a role of c-Jun in mediating the effect of CRS on anxiety and colitis, it is still unclear why they set c-Jun, rather than other molecules associated with neuronal activity, as a research focus in the current study. This is important and the authors should give reasonable explanation in the introduction.
2. The data presented in this manuscript suggested that the c-Jun in AgRP+ cells has a role in stress-induced increase of anxiety-like behavior and colitis susceptibility, and such effect appears to be achieved through altering the secretion of Thbs1. It should be at least discussed how the altered AgRP neuronal activity may cause the alteration of the production and/or secretion of Thbs1.
3. Is the AgRP cell only involved in the stress-induced co-occurrence of anxiety and colitis? Or also

involved in stress-induced anxiety without colitis?? Answering this question may help to understand whether AgRP cells are important for the co-occurrence of anxiety and colitis, or are only required for stress-induced anxiety or colitis per se.

4. It's worth testing whether or not in the c-Jun Δ AgRP mice, the anxiety-like behavior was also altered.

Minor

1. Title: better to add the word "increase of" between "induced" and "anxiety" to improve the readability;

2. Line 22, what the neural mechanisms refer to? Association between psychiatry disorders and IBD? Better to clearly specify.

3. Lines 26 and 101: similar to the title,

4. Line 110: Chemogenetic, not chemogenetically;

5. Deleted "a:"

6. Line 125: enables what/who?

Dear Dr Guo,

Your manuscript entitled "Inhibition of c-Jun in AgRP neurons mediates stress-induced anxiety and colitis susceptibility" has now been seen by 3 referees. You will see from their comments below that while they find your work of considerable interest, some important points are raised. We are interested in the possibility of publishing your study in Communications Biology, but would like to consider your response to these concerns in the form of a revised manuscript before we make a final decision on publication.

We therefore invite you to revise and resubmit your manuscript, taking into account the points raised. In particular, please address all the discussion points raised by the 3 reviewers, taking particular care to expand on the Methods as outlined by Reviewer #1 and #2 and also to specifically address the point raised by R3 regarding the specificity of ArRP activity to colitis.

Please highlight all changes in the manuscript text file.

We are committed to providing a fair and constructive peer-review process. Do not hesitate to contact us if you wish to discuss the revision in more detail or if there are specific requests from the reviewers that you believe are technically impossible or unlikely to yield a meaningful outcome.

At the same time, we ask that you ensure your manuscript complies with our editorial policies. Please see our revision file checklist for guidance on formatting the manuscript and complying with our policies. You will also find guidelines for replying to the referees' comments. You may also wish to review our formatting guidelines for final submissions here.

Please use the following link to submit your revised manuscript, point-by-point response to the referees' comments (which should be in a separate document to the cover letter) and any additional files:

<https://mts-commsbio.nature.com/cgi-bin/main.plex?el=A4Cx7FRu2A3BAsT3I1A9ftdZvrubmyBDn8ZitATQ7L9nQZ>

When submitting the revised version of your manuscript, please pay close attention to our Digital Image Integrity Guidelines.

We would expect revisions of this nature to take around three months, but appreciate that every situation is unique. We look forward to receiving your revised manuscript when it is ready, and will not enforce a hard deadline on this revision.

Please do not hesitate to contact me if you have any questions or would like to discuss these revisions further. We look forward to seeing the revised manuscript and thank you for the opportunity to review your work.

Best regards,

Joao Manuel de Sousa Valente, PhD
Associate Editor
Communications Biology

Referee expertise:

Referee #1: neurocircuits, behavior, and metabolism

Referee #2: nutrition + neurobiology

Referee #3: chronic stress-induced anxiety

Reviewers' comments:

Reviewer #1 (Remarks to the Author):

This paper described interesting observations on AgRP neuron action in control anxiety-related behaviors and the susceptibility to colitis. Multiple approaches including chemogenic manipulations of AgRP neurons as well animal models with c-Jun specific AgRP KO were used to test whether AgRP neuron activity is sufficient and required for the susceptibility to colitis. A series of strong datasets support that AgRP neurons are inhibited by stress, which in turn increases the susceptibility to colitis induced by DSS. Although the results on anxiety-like were not novel, the impact of colitis seems to be completely novel. The experiments were appropriately designed and analyzed. However, some issues were noted during the review.

1) While it is known the intestinal lumen can undergo dynamic turnover, the effect on the colon length seems to be rather surprising as the results showed that over 1 week of treatments exhibited a dramatic change in the length up to 2cm. This set of data may be benefited from further considerations. In addition, the baseline length in controls appears to be very variable, e.g. Fig. 3E the control length was shown to be about 8.5cm while the control in other experiments were shown to be about 6.5cm. At the minimum, the authors should discuss about this including the potential underlying mechanism responsible for this dramatic change in length.

Our response:

Thanks for the reviewer's valuable suggestion. Studies have reported that DSS is directly toxic against colonic epithelial cells of the basal crypts (Wirtz et al., Nat Protoc 2007). The colon shows shortening after acute DSS stimulation. Our findings are consistent with the other previously reported results. For example, Ravindran et al reported that after 7 days of acute DSS administration, *Gcn2*^{-/-} mice exhibited enhanced colon shortening of up to 2cm compared with the control groups (Ravindran et al., Nature 2016). Wei et al reported that DSS induction significantly shortened the colon length (almost 2 cm), which was further aggravated by high-fat diet (Wei et al., ACS Nano 2020). In another study, the interleukin 2 (IL2)-caspase 3 chimeric protein could ameliorate DSS-induced shortening of the colon and restored it by approximately 2cm (Shteingart et al., Gut 2009). These results indicate that the colon exhibits a dramatic change in length after acute DSS stimulation.

Second, we are sorry for the confusion. In fact, the control groups in Figure 1E, Figure 2E, Figure 4F, Supplementary Figure 1F, Supplementary Figure 3C and Supplementary Figure 6I were stimulated by DSS and colon length shown to be approximately 6.5 cm. Whereas, the control group in

Figure 3E did not experience DSS administration. Different treatments in controls caused different colon lengths, while the same treatments in controls resulted in approximately the same colon length.

2) Since acute AgRP neuron activity changes are known to alter feeding and drinking, the potential different amount of DSS intake due to changes in drinking behavior on the observed outcome needs to be considered.

Our response:

We thank the reviewer for the question. As mentioned by the reviewer, acute activation of AgRP neurons leads to changes in food and water intake (Burnett et al., Neuron 2016). To answer the question, we detected DSS consumption during the activation of AgRP neurons with or without stress. As expected, mice with activated neuronal activity of AgRP exhibited more DSS intake during 7 days of DSS stimulation (Figure S1). These results are consistent with those of a previous report showed that activation of AgRP neurons stimulates water intake in the presence of food (Burnett et al., Neuron 2016). Interestingly, the signs of colitis were not worsened by increased DSS intake.

This information has been added to Results (page 6), Supplementary information (Supplementary Figure 2f) in our revised manuscript.

Figure S1. DSS intake during activation of AgRP neurons. Studies were performed using - hM3Dq mice and + hM3Dq mice under the treatment of 3% DSS in drinking water for 7 days to induce acute colitis with (+ CRS) or without (- CRS) stress, simultaneously receiving CNO injections every 12 h per day. Values are expressed as means \pm SEM ($n = 5-6$ per group), with individual data points. Data were analyzed using two-way analysis of variance, followed by Tukey's multiple comparisons test. $*P < 0.05$.

3) In Fig. 4, was the proteomic study done with or without the DSS treatment? What about the effect of DSS treatments alone on THBS1?

Our response:

We thank the reviewer for the questions. The proteomic study was conducted with DSS treatment. As described in the legend for Figure 4, proteomics profiling studies were using - c-Jun - CRS mice, - c-Jun + CRS mice and + c-Jun + CRS mice with 3% DSS for 7 days. Serum was collected after DSS stimulation.

To answer the second question, we examined the levels of serum THBS1 after 6 days of 3% DSS insult and found that THBS1 levels were decreased compared with those in the control group (Figure S2). These results are consistent with the therapeutic effects of THBS1 in inflammation (Gutierrez et al., World J Gastroenterol 2015).

This information has been added to Results (page 10), Supplementary information (Supplementary Figure 8d) in our revised manuscript.

Figure S2. Serum THBS1 levels. Study was conducted using 20- to 22-week-old WT mice with or without 3% DSS administration for 6 days. The THBS1 levels were measured using an ELISA kit (Elabscience, Wuhan, China) according to the manufacturer's recommendations. Data are expressed as the mean \pm SEM (n = 4-6 per group, as indicated), with individual data points. * P <0.05.

Reviewer #2 (Remarks to the Author):

This study investigated the roles of c-Jun in hypothalamic AgRP neurons in stress-induced anxiety and colitis susceptibility. They used chronic restraint stress (CRS) to generate increased anxiety and colitis susceptibility as a condition of psychiatric disorder and inflammatory bowel disease (IBD). Firstly, they compared CRS mice and control mice to show decreased AgRP neuroal activity in CRS group. That suggested these AgRP neurons were inhibited when CRS induced anxiety and colitis susceptibility. To further investigate the functions of AgRP neurons in stress-induced anxiety and colitis susceptibility, they used chemogenic strategy (DREADD-Dq) to specifically activate these AgRP neurons. The activation of AgRP neurons could rescue CRS-increased anxiety behaviors and colitis susceptibility. It suggested the inhibition of AgRP neurons is required to stress-induced anxiety and colitis susceptibility. Then they also used DREADD-Di to specifically inhibit these AgRP neurons. The inhibition of AgRP neurons could induce anxiety behaviors and colitis susceptibility. It suggested the inhibition of AgRP neurons is sufficient to stress-induced anxiety and colitis susceptibility. In addition, they explored the critical role of c-Jun in AgRP neurons in mediating stress-induced anxiety and colitis susceptibility. They confirmed the requirement and sufficiency of c-Jun in AgRP neurons via overexpression and deletion of c-Jun in the regulation of stress-induced anxiety and colitis susceptibility. Finally, they conducted mass spectrometry to screen candidates of secret proteins, and further validated thrombospondin 1 (THBS1) in colitis susceptibility. In summary, this study provide a new neuron-endocrine (c-Jun –AgRP-THBS1) to treat anxiety and IBD. As we know, psychiatric disorders are sex dimorphic. This study only included males, what about females? Did you try in females? Please clarify in discussion.

Our response:

We appreciate the reviewer for highlighting this important issue. To answer the question, we performed experiments on female mice. *c-Jun*^{AgRP} female mice displayed no anxiety-like behaviors as demonstrated by the open field (OF) test and the elevated plus maze (EPM) test (Figure S3 A-B). Moreover, the signs of colitis in *c-Jun*^{AgRP} female mice, including weight loss, rectal bleeding, colon shortening, H&E staining, and mRNA expression of proinflammatory cytokines, did not show significant differences with controls (Figure S3 C-G). These results suggest that the regulation of c-Jun in stress induced anxiety-like behaviors and colitis susceptibility varied in gender.

Previous studies have described sexual dimorphism in the regulation of psychiatric disorders. For example, deleting vasopressin-expressing cells in the paraventricular nucleus increases social investigation only in females and anxiety-related behaviors only in males (Rigney et al., Neuroendocrinology

2021). Moreover, oxytocin receptor interneurons in the medial prefrontal cortex have been shown to mediate prosocial behaviors in female mice and anxiety-related behaviors in male mice (Li et al., Cell 2016). Potential mechanisms underlying the sex differences in anxiety states include neural circuits (Bangasser and Cuarenta, Nat Rev Neurosci 2021), CRF signaling (Brunton et al., Stress 2011), hormones (Zeidan et al., Biol Psychiatry 2011), etc. The detailed mechanism needs to be further investigated.

Sex differences have also been found in the incidence and progression of inflammatory bowel disease (Shah et al., Gastroenterology 2018). The sex hormones, including estrogen, progesterone, and androgen, might account for the pathogenesis of sexual dimorphism in IBD (Xu et al., Inflamm Bowel Dis 2022). Studies have shown that estrogen signaling is important to maintain epithelial homeostasis in a sex- and age-dependent manner (Jacenik et al., Int J Mol Sci 2019). Specifically, estrogen receptor b (ERb) expression in female mice could protect against DSS colitis, whereas it fails to protect male mice (Goodman et al., Cell Mol Gastroenterol Hepatol 2018). Supplementation with estradiol in female mice has been shown to partially protect against DSS- induced colitis (Babickova et al., Inflammation 2015).

In conclusion, hormones may be responsible for the phenotypic differences. The underlying mechanisms need to be further investigated.

This information has been added to Discussion (page 14) in our revised manuscript.

Figure S3. Deletion of c-Jun in AgRP neurons has no effect on anxiety-like behaviors and DSS-induced colitis in female mice. (A) Representative tracks and statistical results in OF test. (B) Representative tracks and statistics in EPM test. (C) Percentage of body weight loss. (D) Scores of diarrhea. (E) Gross morphology and length of the colon. (F) H&E staining and histological scores of the colon tissues. Scale bar, 110 μ m. (G) qRT-PCR analysis of mRNA expression of inflammatory cytokines (*Il6*, *Il1b*, *Il12*, and *Tgfb*) in the distal colonic tissues. Studies for A-B were conducted using 22- to 24-week-old control female mice (*c-Jun*^{loxp/loxp}) or female mice with c-Jun deletion in AgRP neurons (*c-Jun* ^{Δ AgRP}). C-G were performed in *c-Jun*^{loxp/loxp} and *c-Jun* ^{Δ AgRP} female mice with (+ DSS) or without (- DSS) 3% DSS administration for 7 days to induce acute colitis. Values are expressed as means \pm SEM (n = 3-9 per group), with individual data points. Data were analyzed using two-tailed unpaired Student's t test (A-B). Data were analyzed using two-way analysis of variance, followed by Tukey's multiple comparisons test (C-G). **P* < 0.05.

1 What is "pair-fed experiments"? You'd better to add the method.

Our response:

We thank the reviewer for pointing out this issue. We have described the pair-feeding experiments in the legend of Supplementary Figure 3. As suggested, we have refined the descriptions and added it to methods sections.

Pair-feeding (PF) experiments were performed as previously described (Herman et al., Nature 2016). In brief, mice were receiving AAV expressing mCherry (- hM3Dq) or hM3Dq (+ hM3Dq) and individually housed. Food intake was recorded during 14 days of stress. Subsequently, the - hM3Dq and + hM3Dq mice were randomly paired. The PF experiment was conducted during CNO injection. The PF groups (+ hM3Dq + CRS + PF) were given an equivalent amount of food consumed the previous day by its paired partner (- hM3Dq + CRS groups).

The detailed methods have been added to Materials and Methods (page 18) in the revised manuscript.

2 How to treat THBS1? Method? Dose?

Our response:

We thank the reviewer for the questions. The mice were provided ad libitum access to food and water and treated based on body weight. According to related studies (Bai et al., EBioMedicine 2020), mice were subcutaneously bilaterally injected with recombinant human thrombospondin 1 (THBS1) protein (0.5 mg/kg per day; Novoprotein Scientific Inc., Shanghai, China) or vehicle (phosphate-buffered saline) every day for 6 days.

The detailed methods have been added to Materials and Methods (page 16) in the revised manuscript.

3 Main figures:

Figure 1B, bar graph, what about difference and significance of - hM3Dq + CRS and + hM3Dq + CRS?

Our response:

We thank the reviewer for the suggestion. We compared the differences between the two groups. The p value in the open arm entries between - hM3Dq + CRS and + hM3Dq + CRS was 0.044, and that in the open arm time between - hM3Dq + CRS and + hM3Dq + CRS was 0.198. We apologize for the miss of significance and have added it in the revised manuscript.

Figure 4D-H, did you do $c\text{-Jun}^{\text{loxp/loxp}}$ + Thbs1? If Thbs1 can improve colitis susceptibility in control group?

Our response:

We thank the reviewer for the suggestions. We did not analyze $c\text{-Jun}^{\text{loxp/loxp}}$ + THBS1 for two reasons: 1) we aimed to investigate whether THBS1 might mediate the effect of central c-Jun changes on susceptibility to colitis, therefore, the experiments were focused on $c\text{-Jun}^{\Delta\text{AgRP}}$ mice; 2) the effect of THBS1 supplementation have been reported in control mice. As reported, injection of THBS1 mimicking peptide, ABT-510, in WT mice alleviates the weight loss caused by DSS (Punekar et al., Pathobiology 2008). Besides, another THBS1 mimicking peptide, ABT-898, is also shown to alleviate DSS-induced colitis in WT mice, demonstrated by plasma IL-6 levels and H&E staining (Gutierrez LS et al. World J Gastroenterol 2015). Moreover, the 3TSR treatment, which includes three type 1 repeats of the THBS1 is effective in attenuating the inflammatory response to DSS injury (Lopez-Dee et al., PLOS one 2012). These results indicate that THBS1 might inhibit inflammation in $c\text{-Jun}^{\text{loxp/loxp}}$ mice under DSS treatment.

This information has been added to Discussion (page 13) in our revised manuscript.

Figure 4, you only presented colitis susceptibility related. What about anxiety related: open field and elevated plus maze? Thbs1 involved in anxiety-like behavior or not?

Our response:

We thank the reviewer for the suggestions. As suggested, we examined the anxiety-like behaviors in mice after THBS1 treatment. As shown in Figure S4, the anxiety-like behaviors in $c\text{-Jun}^{\Delta\text{AgRP}}$ mice, as demonstrated by the significantly reduced central time and central travel distance in the OF test,

and the shorter time and fewer entries to the open arms in the EPM test, were not relieved after THBS1 replenished (Figure S4 A-B). These results indicate that THBS1 might act downstream of c-Jun in regulation of gut inflammation rather than acting on the brain.

Studies have shown that serum levels of THBS1 are decreased in depressed female patients (Okada-Tsuchioka et al., *Neuropsychopharmacol Rep* 2020). However, few studies have focused on the effect of THBS1 on anxiety, which needs to be further investigated.

This information has been added to Supplementary information (Supplementary Figure 8e-f) and Discussion (page 13) in our revised manuscript.

Figure S4. Supplementary of THBS1 has no effect on anxiety-like behaviors in *c-Jun^{ΔAgRP}* mice. (A) Statistical results in OF test. (B) Statistics in EPM test. Studies were conducted using 22- to 24-week-old *c-Jun^{loxp/loxp}* mice and *c-Jun^{ΔAgRP}* mice with (+ thbs1) or without (- thbs1) THBS1 supplementary. Behavioral tests were performed 30 min after single THBS1 injection. Values are expressed as means \pm SEM (n = 6-8 per group), with individual data points. Data were analyzed using one-way analysis of variance, followed by Tukey's multiple comparisons test. * $P < 0.05$.

4 Supplementary figures:

Supplementary F 2B, when received CNO? Please describe details.

Our response:

We thank the reviewer for the suggestion. As shown in the legend of Figure 1, studies were conducted using 10- to 12-week-old AgRP-Cre mice receiving AAV expressing mCherry (- hM3Dq) or hM3Dq (+ hM3Dq), all mice experienced unstressed (- CRS) or stressed (+ CRS) treatment for 14 days. CNO was administered 30 minutes before the behavioral test on day 15 (OF test) and day 16 (EPM test). We have improved the flow chart of the experimental timeline in Supplementary Figure 2B to make it easy to understand.

This information has been added to Supplementary information (Supplementary Figure 2b) and Materials and Methods (page 18) in our revised manuscript.

Supplementary F 2D, food intake, dark cycle or light cycle? Feed or fasting? Please describe details.

Our response:

We thank the reviewer for the suggestion. The food intake was measured in the light cycle based on the feed condition. Food intake was assessed 30 minutes after CNO injection. According to a previously published paper, even in calorically replete state, the acute activation of AgRP neurons, via clozapine-N-oxide administration, resulted in voracious feeding over the first 2 h (Krashes et al., Cell Metab 2013). Additionally, our results were consistent with a previous study (Krashes et al., 2011), which showed a significant increase in food intake in 30 minutes after CNO injection.

This information has been added to Materials and Methods (page 16) in our revised manuscript.

Supplementary F 4, when inhibiting AgRP neurons, why the timeline of OFT, EPM and 3% DSS is not consistent to activation of AgRP and c-Jun?

Our response:

We thank the reviewer for the valuable suggestion. First, activation of AgRP neurons enables reversal of anxiety-related behaviors and colitis susceptibility induced by 14 days of CRS, while inhibition of AgRP neurons mimics these effects when performed without stress. Second, we also explored the time duration for which the effect of inhibition of AgRP works. According to the pre-test results, the effect was not obvious in the observed phenotype after inhibition of AgRP neurons for 14 days (Figure S5 A-G). We speculate that the function of AgRP is more effective under stress when its activity is inhibited, whereas inhibition under physiological conditions might take longer time to work.

Figure S5. Effect of inhibition of AgRP neurons for 14 days. (A) Representative tracks and statistical results in OF test. (B) Representative tracks and statistics in EPM test. (C) Percentage of body weight loss. (D) Scores of diarrhea. (E) Gross morphology and length of the colon. (F) H&E staining and histological scores of the colon tissues. Scale bar, 110 μ m. (G) qRT-PCR analysis of mRNA expression of inflammatory cytokines (*Il6*, *Il1b*, *Il12*, and *Tgfb*) in the distal colon tissues. Studies for A-B were conducted using 10- to 12-week-old AgRP-Cre mice receiving AAV expressing mCherry (- hM4Di) or hM4Di (+ hM4Di), all mice received CNO injections every 12 h per day for 14 days. Behavioral tests were performed 30 min after single CNO injection on day 15 (A) and day 16 (B). C-G were performed using - hM4Di mice and + hM4Di mice with 3% DSS in drinking water for 7 days to induce acute colitis after 14 days of CNO injections. Values are expressed as means \pm SEM (n = 5-8 per group), with individual data points. Data were analyzed using two-way analysis of variance with Bonferroni's multiple comparisons test (C-D). Data were analyzed using two-tailed unpaired Student's t test (A-B, E-G). * P < 0.05.

Supplementary F 5, what about the expression of c-Jun? Please add the staining of c-Jun.

Our response:

We thank the reviewer for the suggestion. The staining of c-Jun was shown in Supplementary Figure 6c and Supplementary Figure 7a.

Supplementary F 7, you used AgRP-Cre-Ai9 as control in “A”, but c-Jun^{loxp/loxp} as control in “B” and all other experiments about c-Jun. why not use c-Jun^{loxp/loxp} as control also in “A”?

Our response:

We are sorry for the confusion. To characterize the changes of signaling in AgRP neurons or to evaluate the efficiency of AgRP-specific c-Jun deletion, we used the Ai9 reporter system which is Cre-driver lines for cell-type-specific genetic manipulation (Madisen et al., Nat Neurosci 2010). In Supplementary Figure 7a, in order to visualize the changes of c-Jun expression in the AgRP protein-expressing neurons, we mating Ai9 reporter mice with *c-Jun*^{ΔAgRP} transgenic mice to obtain *c-Jun*^{ΔAgRP}-Ai9 mice, and mating Ai9 reporter mice with AgRP-irs-Cre mice to obtain AgRP-Ai9 mice as controls. Other experiments were carried out in transgenic knock out mice (*c-Jun*^{ΔAgRP}) and their littermates (*c-Jun*^{loxp/loxp}).

This information has been added to Materials and Methods (page 16) in our revised manuscript.

5 The writing is not good and not easy reading.

1) There are too many long-complicated sentences which make it hard to read and understand. Please break down these long sentences.

2) Please make sentences direct and concise

Edits eg.

Page 3, line 52 and 53, there are three “of” in one sentence. Please make simple.

Page 3, line 60-64, “Unlike the POMC that food-seeking behavior” I can get your point, however the writing is not smooth. Please edit to make reading easily.

Page 4, line 68, could delete “either”; line 69, could change “and plays a role in the activation of” to “to activate”.

Page 8, line 153, change “this possibility” to “it”.

Page 9, line 174, like “sera”, in this manuscript you sometimes use “sera”, sometimes “serum”. I suggest only “serum”. That will be easy reading.

Line 356-376, please reorganize words.

Our response:

We appreciate the reviewer's comprehensive of our manuscript. As suggested, we have rewritten these sentences as follows:

(1) Line 52-53, we have rewritten the sentence to be as "The hypothalamus is an important neural control center for regulating stress response, consisting of several important nuclei."

(2) Line 60-64, we have rewritten the sentence to be as "The AgRP expressing cells are limited to ARC and its activity is influenced by stress. The AgRP neuronal activity controls diverse physiological processes, including feeding, pain sensation and food-seeking behavior."

(3) Line 68,69 and 153, we have modified the sentences according to the suggestions.

(4) We have changed "sera" to "serum" throughout the manuscript.

(5) We have reorganized words in line 356-376.

As suggested, we have asked the professional English speaker to edit our manuscript to avoid the mistakes in grammar and makes it easy to read.

Reviewer #3 (Remarks to the Author):

In this manuscript, Jiao et al systematically analyzed the neuronal and molecular mechanism underlying the co-occurrence of stress-induced increase in anxiety and colitis susceptibility. They found that CRS markedly decrease the activity of AgRP cells with a concomitant inhibition of c-Jun in this particular cell type. More importantly, artificial regulation of the AgRP cell activity or the c-jun expression was sufficient to mimic /antagonize the effect of CRS. Finally, they showed that the altered Thbs1 secretion may mediating some of the roles of AgRP cells on stress effect. Overall, the data presented herein help to expand our understanding of the neuronal underpinning of stress-induced occurrence of anxiety and colitis. However, there are also some important issues to be considered.

Major:

1. Although the authors have provided evidence to show a role of c-Jun in mediating the effect of CRS on anxiety and colitis, it is still unclear why they set c-Jun, rather than other molecules associated with neuronal activity, as a research focus in the current study. This is important and the authors should give reasonable explanation in the introduction.

Our response:

We thank the reviewer for the valuable suggestion. In addition to influencing neuronal activity, we chose *c-Jun* as a target for two other reasons.

(1) Studies have indicated that c-Jun is involved in a variety of stress responses. For example, a study showed that c-Jun protects cells from stress-

induced apoptosis (Wisdom et al., EMBO J 1999). Another study reported that c-Jun is required for adaptive cardiac hypertrophy in stress-imposed maladaptive remodeling of the heart model (Windak et al., PLoS One 2013). The expression of c-Fos and c-Jun in the hypothalamus is regulated by exhaustive exercise stress (Hong et al., Mol Med Rep 2014). Besides, chronic isolation stress results in decreased c-Jun phosphorylation expression in the prefrontal cortex (Filipovic et al., J Neural Transm (Vienna) 2012). Considering that c-Jun is an immediate early gene that can quickly respond to stress, we chose c-Jun as the target of interest to treat stress-induced efforts.

(2) Several studies have suggested that c-Jun is closely related to brain-associated disease. Mice lacking c-Jun develop severe defects in axonal response after transection of the facial nerve (Raivich et al., Neuron. 2004). Inhibition of c-Jun could attenuate axotomy-induced dopamine neurons death (Crocker et al., Proc Natl Acad Sci U S A. 2001). Current evidence has revealed that activator protein-1 (formed by c-Fos and c-Jun) is activated in the brain by chronic antidepressant treatment and controls the p11-dependent antidepressant response (Chottekalapanda et al., Mol Psychiatry. 2020). These studies suggest that c-Jun may be associated with psychiatric disorders.

Studies have shown that c-Jun is highly expressed in the arcuate nucleus (Herdegen et al., J Comp Neurol. 1995), where the AgRP neurons are enriched, suggesting that it may be involved in the regulation of some important functions in the AgRP neurons.

This information has been added to Introduction (page 4) in our revised manuscript.

2.The data presented in this manuscript suggested that the c-Jun in AgrP+ cells has a role in stress-induced increase of anxiety-like behavior and colitis susceptibility, and such effect appears to be achieved through altering the secretion of Thbs1. It should be at least discussed how the altered Agrp neuronal activity may cause the alteration of the production and/or secretion of Thbs1.

Our response:

We thank the reviewer for the helpful suggestion. We have explained the connections as follows:

(1) Studies have shown that THBS1 can be secreted by astrocytes and promotes CNS synaptogenesis (Christopherson et al., Cell. 2005). Consistently, other studies have revealed that neuron-to-astrocyte communication is established when the physiological state changes. The activation of AgRP neurons can promote changes in neighboring astrocytes

by releasing the inhibitory neurotransmitter GABA (Varela et al., J Clin Invest. 2021). Therefore, we speculate that altered AgRP activity may stimulate the neighboring astrocytes, which in turn causes the secretion of THBS1. This issue remains to be determined.

(2) Several lines of evidence illustrate that the vagus nerve provides parasympathetic innervation to the gastrointestinal tract, coordinating the interactions between central and peripheral, which exerts influence on the development of inflammatory bowel disease (Ghia et al., Gastroenterology. 2006; Ghia et al., J Clin Invest. 2008). Therefore, changes in central nervous system may affect the gut through the vagus nerve, since AgRP neurons show a close connection with vagal (Goldstein et al., Cell Metab. 2021; Strembitska et al., Nat Commun. 2022). Besides, intestinal epithelial cells can produce THBS1 and regulate intestinal inflammatory responses through modulating Mo properties (Fang et al., J Innate Immun. 2015). These studies indicate the possibility of a connection between the neuron and the secretory protein.

This information has been added to Discussion (page 13-14) in our revised manuscript.

3. Is the AgRP cell only involved in the stress-induced co-occurrence of anxiety and colitis? Or also involved in stress-induced anxiety without colitis?? Answering this question may help to understand whether AgRP cells are important for the co-occurrence of anxiety and colitis, or are only required for stress-induced anxiety or colitis per se.

Our response:

We thank the reviewer for the valuable suggestion. It is difficult to figure out whether AgRP neurons are crucial for anxiety and colitis susceptibility comorbidity or are only involved in one disease, as most previous studies have focused only on one condition. Studies have shown that AgRP neurons are involved in chronic unpredictable stress mediated depression-related behaviors and engaged in high-fat diet -induced anxiety and depression (Fang et al., Mol Psychiatry. 2021; Xia et al., Mol Psychiatry. 2021). AgRP neurons are also involved in fasting-induced anxiolytic effects (Li et al., Transl Psychiatry. 2019). However, these studies did not focus on intestinal inflammation.

In fact, in our study, AgRP neurons were involved in stress-induced colitis susceptibility rather than spontaneous colitis, implying that the pathology of colitis requires conditions of DSS stimulation. We used a 14-day restraint model and followed DSS administration to certify that AgRP neurons are important in the stress-induced co-occurrence of anxiety and colitis

susceptibility. More importantly, we aimed to demonstrate the influence of AgRP on stress-induced colitis susceptibility as few studies have focused on this aspect. However, this does not imply that changes in AgRP neurons always accompany these two diseases.

This information has been added to Discussion (page 11-12) in our revised manuscript.

4. It's worth testing whether or not in the *c-Jun*^{ΔAgRP} mice, the anxiety-like behavior was also altered.

Our response:

We thank the reviewer for the suggestion. Deletion of c-Jun in AgRP neurons facilitates anxiety-like behaviors as shown in Figure 3A-B.

Minor

1. Title: better to add the word "increase of" between "induced" and "anxiety" to improve the readability;

Our response:

We thank the reviewer for the suggestion. We have rewritten the title to make it more easily readable.

2. Line 22, what the neural mechanisms refer to? Association between psychiatry disorders and IBD? Better to clearly specify.

Our response:

We apologize for the confusion. We have rewritten the sentence to make it more specific.

3. Lines 26 and 101: similar to the title,

Our response:

We appreciate the reviewer's careful reading of our manuscript. As suggested, changes have been made in the revised manuscript.

4. Line 110: Chemogenetic, not cehmogenetically;

Our response:

We appreciate the reviewer's comprehensive review of our manuscript. We have changed "chemogenetically" to "chemogenetic" in the revised manuscript.

5. Deleted "a:

Our response:

We apologize for this error. We have deleted “a” in the sentence.

6. Line 125: enables what/who?

Our response:

We apologize for the confusion. We have rewritten the sentence to make it clearer in the revised manuscript.

** See the Nature Portfolio author and referees' website at www.nature.com/authors for information about policies, services and author benefits

Communications Biology is committed to improving transparency in authorship. As part of our efforts in this direction, we are now requesting that all authors identified as 'corresponding author' create and link their Open Researcher and Contributor Identifier (ORCID) with their account on the Manuscript Tracking System prior to acceptance. ORCID helps the scientific community achieve unambiguous attribution of all scholarly contributions. You can create and link your ORCID from the home page of the Manuscript Tracking System by clicking on 'Modify my Springer Nature account' and following the instructions in the link below. Please also inform all co-authors that they can add their ORCIDs to their accounts and that they must do so prior to acceptance.

<https://www.springernature.com/gp/researchers/orcid/orcid-for-nature-esearch>

If you experience problems in linking your ORCID, please contact the Platform Support Helpdesk.

This email has been sent through the Springer Nature Tracking System NY-610A-NPG&MTSConfidentiality Statement: This e-mail is confidential and subject to copyright. Any unauthorised use or disclosure of its contents is prohibited. If you have received this email in error please notify our Manuscript Tracking System Helpdesk team at <http://platformsupport.nature.com>. Details of the confidentiality and pre-publicity policy may be found here <http://www.nature.com/authors/policies/confidentiality.html> Privacy Policy Update Profile

REVIEWERS' COMMENTS:

Reviewer #1 (Remarks to the Author):

The authors have adequately addressed the concerns raised in the previous round and recommend acceptance.

Reviewer #2 (Remarks to the Author):

My concerns have been addressed very well in the rebuttal letter and revised manuscript. It is a good story.

Reviewer #3 (Remarks to the Author):

The authors have satisfactorily addressed all the concerns I raised. I have no more comments and recommend to publish this exciting work in Communications Biology.